# Clinical predictors of mortality in patients with pseudomonas aeruginosa infection

**Jim Abi Frem[1]◦, George Doumat [1]◦, Jamil Kazma[2], Amal Gharamti[1], Souha S. Kanj[1], Antoine G. Abou Fayad[3,4], Ghassan M. Matar[3,4], Zeina A. Kanafani [1]***

**1** Department of Internal Medicine, American University of Beirut, Beirut, Lebanon, **2** Department of Obstetrics & Gynecology, George Washington University School of Medicine, Washington, District of Columbia, United States of America, **3** Department of Experimental Pathology, Immunology, and Microbiology, American University of Beirut, Beirut, Lebanon, **4** WHO Collaborating Center for Reference and Research on Bacterial Pathogens, American University of Beirut, Beirut, Lebanon

◦ These authors contributed equally to this work.
* zk10@aub.edu.lb

**Data Availability Statement:** All relevant data are within the paper.

## Abstract

### Background

Infections caused by *Pseudomonas aeruginosa* are difficult to treat with a significant cost and burden. In Lebanon, *P. aeruginosa* is one of the most common organisms in ventilator-associated pneumonia (VAP). *P. aeruginosa* has developed widespread resistance to multiple antimicrobial agents such as fluoroquinolones and carbapenems. We aimed at identifying risk factors associated for *P. aeruginosa* infections as well as identifying independent risk factors for developing septic shock and in-hospital mortality.

### Methods

We used a cross-sectional study design where we included patients with documented *P. aeruginosa* cultures who developed an infection after obtaining written consent. Two multi-variable regression models were used to determine independent predictors of septic shock and mortality.

### Results

During the observed period of 30 months 196 patients were recruited. The most common predisposing factor was antibiotic use for more than 48 hours within 30 days (55%). The prevalence of multi-drug resistant (MDR) *P. aeruginosa* was 10%. The strongest predictors of mortality were steroid use (aOR = 3.4), respiratory failure (aOR = 7.3), identified respiratory cultures (aOR = 6.0), malignancy (aOR = 9.8), septic shock (aOR = 18.6), and hemodialysis (aOR = 30.9).

### Conclusion

Understanding resistance patterns and risk factors associated with mortality is crucial to personalize treatment based on risk level and to decrease the emerging threat of antimicrobial resistance.

**Funding:** This study was supported by the Medical Practice Plan, American University of Beirut, in the form of a grant to ZAK [320153].

**Competing interests:** The authors have declared that no competing interests exist.

## Introduction

*Pseudomonas aeruginosa* is an increasingly recognized pathogen worldwide, particularly in hospital-acquired infections (HAI). Invasive infections due to *P. aeruginosa* are often difficult to treat [1, 2], and this is evident in cases of bacteremia, pneumonia, bloodstream infections, and intra-abdominal infections [3]. *P. aeruginosa* is a leading pathogen in ventilator-associated pneumonia (VAP) globally [4]. In Lebanon, a recent study determined *P. aeruginosa* to be the second most common pathogen isolated in VAP [5]. *P. aeruginosa* infections are associated with a significant burden and cost. For instance, the additional cost of a single case of *P. aeruginosa* pneumonia has been estimated to be 19,000 Euros in a German teaching hospital [1]. *P. aeruginosa* easily acquires resistance to commonly used antimicrobial agents [6, 7]. In Lebanon, carbapenem resistance among *P. aeruginosa* isolates is estimated to be 28% [8]. This is consistent with regional data, with carbapenem resistance reaching 56% in Libya, 51% in Egypt, and 93% in Jordan [9].

There are established clinical risk factors associated with *P. aeruginosa* infections, such as previous hospital admission, history of antibiotic use, and ventilator use [6]. In a recent study, prolonged intensive care unit (ICU) stay was also shown to predispose to *P. aeruginosa* infections, particularly with resistant strains [3]. As pseudomonal infections are often associated with high mortality rates, studies have examined risk factors for mortality and have found independent predictors, such as multi-drug resistance, neutropenia, increased Pitt bacteremia score, and delay in therapy [10].

The aim of this study is to quantify the mortality risk in hospitalized patients with *P. aeruginosa* infections by developing a risk score. This would allow clinicians to gauge the individual mortality risk for their patients based on easily obtainable clinical data. Such patients could benefit from more aggressive treatment and close follow-up.

## Materials and methods

### Study design

This prospective chart review study from 2017 to 2020 involving human participants was in accordance with the ethical standards of the institutional and national research committee and with the 1964 Helsinki Declaration and its later amendments or comparable ethical standards. The Institutional Review Board of the American University of Beirut, a tertiary healthcare institution in Lebanon approved this study. Written consents were obtained. Study subjects were hospitalized adult patients with documented *P. aeruginosa* infections for which treatment was initiated. We classified these hospitalized patients based on where the infection was acquired (community-acquired, hospital-acquired, or healthcare-associated). Infections were considered community-acquired if they manifested outside the hospital or were diagnosed within 48 hours of admission without any healthcare encounter in the past 30 days. Hospital-acquired infection was defined as an infection that manifested 48 hours or more after hospital admission or within 7 days of discharge. Healthcare-associated infection was defined as an infection occurring in patients receiving home and/or ambulatory intravenous therapy, chemotherapy, hemodialysis, wound care, specialized nursing care, or who had attended a hospital clinic within the last 30 days; patients hospitalized in an acute care hospital for $\geq 2$ days within the last 90 days; and those residing in a nursing home or long-term care facility. Patients were included in the study only once, taking into consideration the first *P. aeruginosa* infection within the study period. Recurrent *P. aeruginosa* infections and subsequent hospitalizations were accounted for as complications. Patients who were deemed to be colonized with *P. aeruginosa* were excluded. We relied on the assessment of the infectious diseases consult

service in determining infection vs. colonization. In general, patients were considered infected if they displayed typical infection symptoms, such as fever, and symptoms related to the infection type. In case of any uncertainty, another study researcher would have reassessed the eligibility of the patient.

## Subject enrollment

The study coordinators were notified daily by the clinical microbiology laboratory of clinical specimens growing *P. aeruginosa*. After obtaining written consent, the medical records of study subjects were reviewed, collecting data on demographics, comorbidities, previous antibiotic use, as well as treatment, complications, and outcome. Based on the Infectious Diseases Society of America (IDSA), multi-drug resistance (MDR) is defined as *P. aeruginosa* not susceptible to at least one antibiotic in at least three antibiotic classes for which *P. aeruginosa* susceptibility is generally expected: penicillins, cephalosporins, fluoroquinolones, aminoglycosides, and carbapenems [11]. Difficult-to-treat (DTR) resistance is defined as *P. aeruginosa* exhibiting non-susceptibility to all of the following: piperacillin-tazobactam, ceftazidime, cefepime, aztreonam, meropenem, imipenem-cilastatin, ciprofloxacin, and levofloxacin [12].

## Statistical analysis

The independent samples T-test was used to compare continuous variables and Pearson's Chi-Square test was used to compare categorical variables. Backward stepwise multivariable logistic regression was used for independent association testing while controlling for potential confounders. The logistic model included all variables with a p-value of 0.2 or less obtained on bivariable analysis. The variables that were significant in the mortality model were used to create a clinical scoring system to estimate the mortality due to *P. aeruginosa* infections. Each variable was assigned several points based on its respective adjusted odds ratio obtained on multivariable analysis. Receiver operating characteristic (ROC) analysis was used to determine the best cutoff score. The data was analyzed using SPSS® for Windows, version 18 (SPSS Inc., Chicago, IL, USA).

## Results

During the observed period of 30 months, 196 patients were recruited for the study. Table 1 shows the demographic characteristics of these patients. There was a slight male predominance among the study subjects (58.7%), and the average age was 65.4 ± 19.0 years. The mean Charlson comorbidity index was relatively low at 5.6 ± 3.2. The most common pre-existing condition was hypoalbuminemia (46.8%) followed by malignancy (39.8%). Infections were hospital-acquired in half of the cases, community acquired in one quarter of the cases, and healthcare associated in the remaining quarter. Among the hospital-acquired infections, 40% were acquired in the medical wards, 32% in the intensive care unit, and 20% in the surgical wards.

Among patients who had received antibiotics within 30 days for more than 48 hours before infection, 74.0% had received anti-pseudomonal antibiotics and 27.6% grew a strain of *P. aeruginosa* that was resistant to the antibiotic received. The results stratified by antibiotic class are summarized in Table 2.

*P. aeruginosa* isolates were recovered most frequently from respiratory specimens (48.0%), followed by urine (20.3%), and blood (7.7%), and in 24.0% of cases, the organism was isolated from various other sources including wounds, abdominal fluid, cerebrospinal fluid, and skin abscesses. In healthcare-associated infections, 10 were from urine, 23 from respiratory specimens, 3 from blood, and 12 from other sources. We did not find any statistical difference in

**Table 1. Demographic characteristics of patients with *Pseudomonas aeruginosa* infections.**

| Characteristic (n = 196) | Value |
|---|---|
| Age, *years* | 65.4 ± 19.0 |
| Male gender | 115/196 (58.7) |
| Hospital stay, *days* | 30.6 ± 41.4 |
| Body mass index*, *kg/m²* | 26.4 ± 6.5 |
| Hypoalbuminemia | 73/156 (46.8) |
| Malignancy | 78/196 (39.8) |
| Chronic pulmonary disease | 60/196 (30.6) |
| Diabetes mellitus | 58/196 (29.6) |
| Renal insufficiency | 40/196 (20.4) |
| Hemodialysis | 9/196 (4.6) |
| Charlson comorbidity index | 5.6 ± 3.2 |
| Steroid therapy within 30 days | 50/196 (25.5) |
| Surgery within 30 days | 46/196 (23.5) |
| Mechanical ventilation within 30 days | 36/196 (18.4) |
| Antibiotic therapy within 24 hours | 55/195 (28.2) |
| Antibiotic therapy for > 48 hours within 30 days | 108/195 (55.4) |

Values represent n/N (%) for categorical variables and mean ± standard deviation for continuous variables.

*Data available for 162 patients only.

the distribution of the types of infection by place of acquisition (hospital, healthcare, or community). Most *P. aeruginosa* isolates were sensitive to 3 or more classes of antibiotics (177/196; MDR 10%). The highest overall proportion of susceptibility was for respiratory isolates (93%) and the lowest was for blood isolates (78%). Overall carbapenem susceptibility was 82% among all isolates. DTR was detected in 6/196 isolates (3.1%).

The most commonly prescribed empiric therapy while awaiting culture results was piperacillin-tazobactam in 33% of cases, followed by meropenem (24%), and ciprofloxacin (14%), ceftazidime (9%), and cefepime (7%). A single dose of amikacin was used as adjunctive treatment in 14% of the cases. Ciprofloxacin was the most used antibiotic for directed therapy (40%), and ceftazidime (18%), followed by levofloxacin (11%), piperacillin-tazobactam (8.5%) and cefepime (7.5%). Average duration of empirical therapy was 7.3 days ± 6.9, and that of directed therapy was 10.6 days ± 8.4. Patients who received dual antipseudomonal therapy had their complications subsequently analyzed and compared to those who received monotherapy. Based on our hospital care practices, patients are generally treated with dual anti-pseudomonal therapy if the organism is multidrug-resistant, and in the setting of severe sepsis, neutropenia with bacteremia, and severe pneumonia in ICU. The results shown in Table 3 clearly indicate that dual antipseudomonal therapy was associated with significantly fewer complications and fewer deaths.

**Table 2. Correlation between antibiotics received for > 48 hours within 30 days and isolates resistant to the same antibiotic class.**

| Antibiotics | No. of patients | No. (%) of resistant isolates |
|---|---|---|
| Cephalosporins | 8 | 4 (50.0) |
| Carbapenems | 39 | 14 (35.9) |
| Fluoroquinolones | 22 | 7 (31.8) |
| Piperacillin-tazobactam | 36 | 4 (11.1) |

**Table 3. Clinical complications reported in patients who received dual antipseudomonal therapy compared to those reported in patients who received monotherapy.**

| Complication | Dual therapy | | OR (95% CI) | p-value |
|---|---|---|---|---|
| | 0–2 days (n = 49) | > 2 days (n = 147) | | |
| Sepsis | 32 (65.3) | 19 (12.9) | 0.2 (0.1–0.4) | < 0.001 |
| Septic shock | 23 (46.9) | 14 (9.5) | 0.2 (0.1–0.5) | < 0.001 |
| Acute kidney injury | 20 (40.8) | 11 (7.5) | 0.3 (0.1–0.7) | 0.002 |
| Progression of infection | 34 (69.4) | 18 (12.2) | 0.2 (0.1–0.5) | < 0.001 |
| Recurrent infection | 29 (59.2) | 12 (8.2) | 0.4 (0.2–0.9) | 0.02 |
| Prolonged hospital stay | 40 (81.6) | 21 (14.3) | 0.2 (0.1–0.4) | < 0.001 |
| ICU admission | 38 (77.5) | 15 (10.2) | 0.4 (0.2–0.8) | 0.009 |
| Respiratory failure | 17 (34.7) | 11 (7.5) | 0.2 (0.1–0.6) | 0.001 |
| Hospital-acquired infection | 37 (75.5) | 18 (12.2) | 0.2 (0.1–0.5) | < 0.001 |
| Death | 26 (53.1) | 12 (8.2) | 0.3 (0.1–0.8) | 0.007 |

OR = odds ratio; CI = confidence interval; ICU = intensive care unit.

The most common complications encountered were the acquisition of a hospital-acquired infection other than the index infection (28.1%), admission to the intensive care unit (27.0%), sepsis (26.0%), recurrent infection (21.2%), and septic shock (18.9%). In-hospital mortality was 19.4% and mortality was attributed to infection in 26.3% of all patients who died. Mortality was highest when the initial source of infection was pneumonia (30.9%), compared to 15.0% for urinary tract infections, and 6.7% for primary bacteremia. Backward stepwise multivariable logistic regression was performed to test independent risk factors for mortality (Table 4). On bivariable analysis, sepsis, acute respiratory distress syndrome (ARDS), and ICU admission were associated with a worse outcome. However, this relationship was no longer significant when we adjusted for confounders. A total of 53/196 (27.0%) patients required ICU admission. With a p-value of 0.011, ICU mortality was 17/55 (30.9%) and non-ICU mortality was 21/196 (17.5%). The baseline variable with the strongest independent association with death was hemodialysis (aOR = 30.9; 95% CI 2.6–173.0). This was followed by malignancy

**Table 4. Bivariable and multivariable analysis of predictors of mortality among patients with _P. aeruginosa_ infection.**

| Risk factor | No. (%) of patients (n = 196) | Unadjusted OR (95% CI) | Adjusted OR* (95% CI) |
|---|---|---|---|
| **Baseline variables** | | | |
| Hemodialysis | 9 (4.6) | 3.6 (0.9–14.0) | 30.9 (2.6–173.0) |
| Malignancy | 78 (39.8) | 4.4 (2.1–9.4) | 9.8 (2.6–37.6) |
| Positive respiratory culture | 88 (44.9) | 4.4 (1.9–9.9) | 6.0 (1.5–24.2) |
| Steroid therapy within 30 days | 50 (25.5) | 3.2 (1.5–6.8) | 3.4 (1.01–11.6) |
| **Complication variables** | | | |
| Septic shock | 37 (18.9) | 11 (4.8–24.9) | 18.6 (3.0–113.0) |
| Respiratory failure | 28 (14.3) | 8.8 (3.7–21.2) | 7.3 (1.6–34.5) |
| Metastatic infection | 14 (7.1) | 3.5 (1.1–10.8) | 0.1 (0.01–0.8) |
| Sepsis | 51 (26.0) | 6.1 (2.9–13.1) | NS |
| ARDS | 12 (6.1) | 4.8 (1.4–15.7) | NS |
| ICU admission | 53 (27.0) | 2.7 (1.3–5.7) | NS |

*Adjusted for all variables with p-value < 0.2 on bivariable analysis.

OR = odds ratio; CI = confidence interval; ARDS = acute respiratory distress syndrome; ICU = intensive care unit.

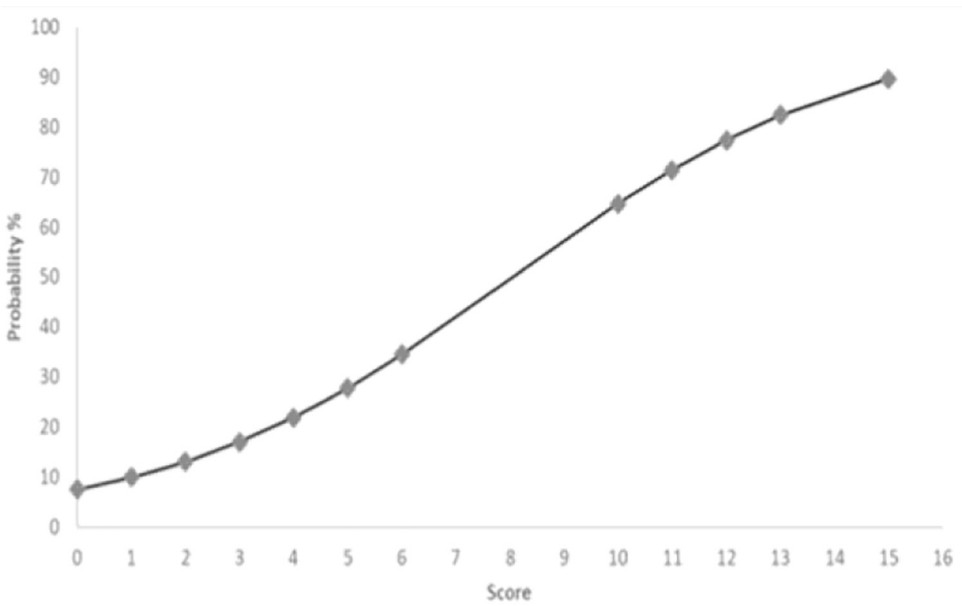

**Fig 1. Association between Pseudomonas mortality score and probability of death.** Pseudomonas mortality score = sum of individual risk factors (1 point for recent steroid use, 2 points for identified respiratory culture, 3 points for malignancy, 10 points for hemodialysis).

(aOR = 9.8; 95% CI 2.6–37.6), positive respiratory culture (aOR = 6.0; 95% CI 1.5–24.2), and recent steroid therapy (aOR = 3.4; 95% CI 1.01–11.6). Other variables, classified as complication variables, were also independently associated with mortality and included septic shock, respiratory failure, and metastatic infection.

Using standard methods [13], a risk of mortality score was generated to assess the likelihood of dying from a *P. aeruginosa* infection. This proposed score was based on the 4 baseline variables. The lowest coefficient variable, recent use of steroids, was assigned 1 point. Isolation of *P. aeruginosa* from respiratory cultures was assigned 2 points, while malignancy was assigned 3 points, and hemodialysis 10 points, because their coefficients were 2, 3, and 10 times that of recent steroid use, respectively. The minimum possible score was 0 and the maximum score possible was 16. The predicted probability of death was 3.8% if no risk factors were present and increased with additional risk factors reaching a maximum of 89.7% at a risk score of 15 (Fig 1). The mean risk score was 2.36 ± 2.38 in the survival group and 5.08 ± 3.36 in the mortality group ($p < 0.001$). The area under the receiver operating characteristic curve was 0.801 ($p < 0.001$). ROC analysis of the mortality score showed that a cutoff of 1.5 points is appropriate to maximize sensitivity (100%) but not specificity (33.5%) in predicting mortality. On the other hand, a cutoff of 2.5 points maximizes both sensitivity (89.2%) and specificity (59.5%).

## Discussion

Predicting the risk of clinical complications is a very important factor in the management of *P. aeruginosa* infections. The accurate prediction of the risk of mortality would help clinicians make specific and individualized management decisions. To our knowledge, this study represents the first prospective cohort study to report the clinical epidemiology of *P. aeruginosa* infections at a large tertiary care center in Lebanon, and the first to create a clinically relevant tool to assess an individual patient's mortality risk.

The morbidity and mortality associated with *P. aeruginosa* infections underscore the importance of hospital infection control efforts, particularly since 50% of the infections in our cohort were acquired in the hospital. However, unlike other studies where most *P. aeruginosa* infections in the hospital are acquired in the intensive care unit (68.7%) [14], most of our nosocomial infections originated in the medical wards.

The ability of *P. aeruginosa* to acquire antimicrobial resistance has been extensively highlighted [15]. In our study, however, the prevalence of MDR *P. aeruginosa* was 10%, and DTR was very low at 3.1%, which is consistent with previous data from our center [8], but lower than what is reported from studies across Europe. In a nationwide Spanish survey, 17.3% of *P. aeruginosa* isolates were extensively-drug resistant and 26.2% were MDR [16], whereas, in France, the prevalence of MDR *P. aeruginosa* was 26.9% [17]. Reports on the prevalence of MDR isolates from the Middle East region also reveal a higher prevalence of MDR isolates. A 1-year retrospective cohort study found a prevalence of 16.3% of *P. aeruginosa* MDR isolates among critically ill patients in Saudi Arabia [18]. Another study from Egypt revealed a prevalence of 66.6% [19].

Almost half of our patients received an antibiotic for more than 48 hours within 30 days of developing an infection with *P. aeruginosa*. It has been previously shown that receiving piperacillin-tazobactam and/or cefepime in the month prior to the index infection is associated with infection with isolates resistant to these antibiotics [20]. In our study, receiving piperacillin-tazobactam was the least associated with developing an infection with an isolate resistant to this drug. However, in 50% of the cases, receiving cephalosporins for more than 48 hours before the index infection predicted the development of resistance to cephalosporins. There is a direct correlation between anti-pseudomonal antimicrobial use and the development of resistance among *P. aeruginosa* to certain classes of antibiotics [21, 22]. Djordjevic et al. studied the effect of cefepime withdrawal on the resistance rates of *P. aeruginosa* [23]. They found that with the withdrawal of cefepime and a decrease in ceftazidime utilization, there was a decrease in the resistance density of *P. aeruginosa* to carbapenems, piperacillin-tazobactam, ceftazidime, and cefepime.

The tendency for *P. aeruginosa* to acquire antimicrobial resistance upon exposure to antibiotics has been one of the drivers for the use of combination therapy. It is proposed that combination therapy can prevent the development of resistance, although definitive evidence is still lacking [24]. Combination therapy also enhances the adequacy of empiric antimicrobial therapy and provides potential synergy between different classes of antibiotics which allows for enhanced bacterial killing [24]. A Spanish study looking at the effect of the extensively drug-resistant (XDR) phenotype on outcome found no effect of the XDR phenotype on 14- or 30-day mortality [25]. However, in the subgroup of patients with high-risk sources, combination therapy significant reduced 14-day mortality (HR 0.56, 95% CI, 0.33–0.93). Similar findings were reported from a recent systematic review, showing that combination therapies for MDR and XDR *P. aeruginosa* matched or outperformed monotherapy, and none were inferior to monotherapies [26]. In our study, all complications were significantly less frequent in the combination therapy arm. This finding, despite the low prevalence of resistance among our isolates, highlights the possibility of more efficient microbial killing with dual antipseudomonal therapy.

The in-hospital mortality rate that we observed (19%) is lower than what has been reported worldwide. For instance, a recent study in China found a 28.4% mortality for *P. aeruginosa* infections [27]. Some of the known mortality risk factors include cardiovascular disease, MDR phenotype, mechanical ventilation, central venous catheter, septic shock, delayed appropriate therapy, chronic obstructive pulmonary disease, neutropenia, and hypoalbuminemia [27–31]. The most important predictor of mortality in our study was hemodialysis, which was

associated with 31-fold increased odds of death among patients with *P. aeruginosa* infections. The cause of this increased mortality is multifactorial. The pharmacokinetics of antimicrobials in patients receiving hemodialysis requires special consideration and dosage adjustments, and therefore, therapeutic concentrations might not be achieved in hemodialysis patients [32, 33]. In addition, immune system dysregulation and impaired innate immunity are observed in dialysis patients, which could jeopardize their immune response to an infection [34, 35]. Another risk factor in our study was malignancy, with a 10-fold increase in odds of death, which can be explained by a longer hospital stay, immunosuppression, and anti-neoplastic chemotherapy [36]. In addition to that, isolation of *P. aeruginosa* from a respiratory source was associated with mortality, which is consistent with the high mortality rates associated with hospital-acquired pneumonias and ventilator-associated pneumonias caused by the organism [3, 4]. The ability of *P. aeruginosa* to induce cell death is critical. Massive lung epithelial cell death due to multiple apoptosis pathways is the hallmark of severe lung injury caused by *P. aeruginosa* [37]. The final independent predictor of mortality was the recent use of steroids, which has a well-established association with a higher risk of bacteremia and mortality [38]. This can be explained by the ability of corticosteroids to suppress phagocytosis, cytokine release, and leukocyte adhesion [39].

In this study, we were interested in developing a mortality risk score that includes baseline variables that would allow the early identification of patients at high risk of death. This risk-stratification process could then be used to guide the management of such patients. The score presented in this study combines the independent risk factors and presents them in a quantitative form to be used with ease in the clinical setting.

This study has several limitations. We recruited each patient only once, and we did not collect data regarding the development of subsequent infection with a resistant pathogen. Therefore, we could not study the effect of monotherapy versus combination therapy on the development of resistance. Even though the individual patient mortality risk score provides a useful tool to stratify patients based on risk, it needs to be validated in larger cohorts before being used in a clinical setting.

## Conclusions

Understanding risk factors associated with *P. aeruginosa* resistance and complications is crucial to decreasing the emerging threat of the organism. Selecting the most suitable empiric treatment based on local resistance patterns as well as individual mortality risk estimations would enable physicians to treat such infections promptly and effectively, to decrease the burden of the disease.

## Author Contributions

**Conceptualization:** George Doumat, Zeina A. Kanafani.

**Data curation:** Jim Abi Frem, George Doumat, Jamil Kazma, Amal Gharamti.

**Formal analysis:** Jamil Kazma.

**Investigation:** Antoine G. Abou Fayad.

**Methodology:** Souha S. Kanj, Antoine G. Abou Fayad, Ghassan M. Matar.

**Resources:** Jim Abi Frem, George Doumat, Jamil Kazma.

**Supervision:** Zeina A. Kanafani.

**Validation:** Zeina A. Kanafani.

**Visualization:** Ghassan M. Matar, Zeina A. Kanafani.

**Writing – original draft:** Jim Abi Frem, George Doumat.

**Writing – review & editing:** Souha S. Kanj, Zeina A. Kanafani.

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
