## [Decision Letter · Decision Letter 0]

2 Jan 2023

PONE-D-22-33603CLINICAL PREDICTORS OF MORTALITY IN PATIENTS WITH PSEUDOMONAS AERUGINOSA INFECTIONPLOS ONE

Dear Dr. Kanafani,

Thank you for submitting your manuscript to PLOS ONE. After careful consideration, we feel that it has merit but does not fully meet PLOS ONE’s publication criteria as it currently stands. Therefore, we invite you to submit a revised version of the manuscript that addresses the points raised during the review process.

We look forward to receiving your revised manuscript.

Kind regards,

Benjamin M. Liu, MBBS, PhD, D(ABMM), MB(ASCP)

Academic Editor

PLOS ONE

Journal Requirements:

Reviewers' comments:

Reviewer's Responses to Questions

**Comments to the Author**

1. Is the manuscript technically sound, and do the data support the conclusions?

Reviewer #1: Yes

Reviewer #2: Yes

2. Has the statistical analysis been performed appropriately and rigorously? 

Reviewer #1: Yes

Reviewer #2: Yes

3. Have the authors made all data underlying the findings in their manuscript fully available?

Reviewer #1: Yes

Reviewer #2: Yes

4. Is the manuscript presented in an intelligible fashion and written in standard English?

Reviewer #1: Yes

Reviewer #2: Yes

5. Review Comments to the Author

Reviewer #1: The overall manuscript is very well written, detailed and well explained. The data is presented well, the literature review is lucid, and the figures are very clear. However, I would like to recommend minor revisions for the manuscript.

1. Abstract --" Pseudomonas aeroginosa" should be in italics.

2. Introduction --" Intra-abdominal" should be hyphenated.

3. Materials and Methods -- Please remove the term " Definitions". Also P. aeroginosa should be in italics.

Reviewer #2: Pseudomonas aeruginosa is a relevant microorganism associated with severe infections, mostly in hospitalized persons. The manuscript is about predictors of mortality in patients with P aeruginosa infections in Lebanon.

The study has an understandable designed but I miss information concerning methodology and data

The study was a cross-sectional one, involving 196 patients from 2017 to 2020 with positive cultures for P aeruginosa who developed an infection.

I wonder how infections were diagnosed as the differential diagnosis between infection and colonization is not always easy. And how many patients with a first colonization became infected?

The authors enroled in the same study patients in hospitalization (50%), outpatients (25%) and health care associated (25%) patients. In the later group, the infection was treated where?

I wonder if this heterogeneity of the group cannot give be responsible for differences from other studies in the literature, concerning for instance prevalence of MDR isolates

I would like to have more information about the type of clinical infection, as this is a clinical study. What about the health-care infections for instance?

In 24% of cases P aeruginosa was not isolated from the blood, respiratory specimes or urine. From where the strains were isolated? At what kind of infections were they associated?

ICU acquired infections were more severe than the others? And how many patients were in the need of IcU admission?

What were the criteria for a dual antipseudomonal therapy?

Finally some remarks

a) Abstract

“Pseudomonas aeruginosa (PSA) is difficult to treat microorganism..”- in spite of “the infection caused by PSA is difficult to treat..”

b) Page 16 - “Similar findings were reported in.” need to be finished the sentence

6. PLOS authors have the option to publish the peer review history of their article (what does this mean?). If published, this will include your full peer review and any attached files.

Reviewer #1: No

Reviewer #2: No

---

## [Author Response · Author response to Decision Letter 0]

11 Jan 2023

Response to Reviewers

Reviewer #1: The overall manuscript is very well written, detailed and well explained. The data is presented well, the literature review is lucid, and the figures are very clear. However, I would like to recommend minor revisions for the manuscript.

1. Abstract --" Pseudomonas aeruginosa" should be in italics.

2. Introduction --" Intra-abdominal" should be hyphenated.

3. Materials and Methods -- Please remove the term " Definitions". Also P. aeruginosa should be in italics.

Thank you for your comments. 

We have made all the proposed edits. 

Reviewer #2: Pseudomonas aeruginosa is a relevant microorganism associated with severe infections, mostly in hospitalized persons. The manuscript is about predictors of mortality in patients with P aeruginosa infections in Lebanon.

The study has an understandable design but I miss information concerning methodology and data

The study was a cross-sectional one, involving 196 patients from 2017 to 2020 with positive cultures for P aeruginosa who developed an infection.

I wonder how infections were diagnosed as the differential diagnosis between infection and colonization is not always easy. And how many patients with a first colonization became infected?

Thank you for pointing this out. There was an official Infectious Diseases consultation on all patients. We relied on the assessment of the ID consult service in determining infection vs. colonization. In general, patients were considered infected if they displayed typical infection symptoms, such as fever, and symptoms related to the infection site. In case of any uncertainty, another study researcher would have reassessed the eligibility of the patient. Unfortunately, we did not collect information about colonization status prior to the index infection. Hence, we cannot tell how many colonized patients later became infected.

The authors enrolled in the same study patients in hospitalization (50%), outpatients (25%) and health care associated (25%) patients. In the later group, the infection was treated where? I wonder if this heterogeneity of the group cannot give be responsible for differences from other studies in the literature, concerning for instance prevalence of MDR isolates

Thank you for your response. To clarify, our patient population consisted only of hospitalized patients. However, we classified these hospitalized based on where the infection was acquired (community-acquired, hospital-acquired, or healthcare-associated). We have clarified this in the Methods section.

I would like to have more information about the type of clinical infection, as this is a clinical study. What about the health-care infections for instance?

The sites of clinical infection are mentioned in the Results section (48% respiratory, 20.3% urine, 7.7% blood, 24% other). In healthcare-associated infections, 10 were from urine, 23 from respiratory specimens, 3 from blood, and 12 from other sources. We did not find any statistical difference in the distribution of sites of infection by place of acquisition (hospital, healthcare, or community). We have added this information to the text.

In 24% of cases P aeruginosa was not isolated from the blood, respiratory specimens or urine. From where the strains were isolated? At what kind of infections were they associated?

Various other sites of infection included wounds, abdominal fluid, CSF, skin abscess. This was added to the Results.

ICU acquired infections were more severe than the others? And how many patients were in the need of ICU admission?

On bivariable analysis (Table 4), we found that ICU admission was associated with higher mortality. However, this relationship was no longer significant when we adjusted for coufounders as evidenced by the regression analysis. We have added this information to the text. A total of 53/196 (27%) patients required ICU admission, which is mentioned in the complications paragraph.

What were the criteria for a dual antipseudomonal therapy?

Thank you for your comment. Since this is a retrospective study, the decision on treatment resided solely with the treating physician. Based on our hospital care practices, however, patients are generally treated with dual anti-pseudomonal therapy if the organism in multidrug-resistant, and in the setting of severe sepsis, neutropenia with bacteremia, and severe pneumonia in ICU.

Finally some remarks

a) Abstract

“Pseudomonas aeruginosa (PSA) is difficult to treat microorganism..”- in spite of “the infection caused by PSA is difficult to treat..”

Thank you. We modified it.

b) Page 16 - “Similar findings were reported in.” need to be finished the sentence

Thank you. We removed the sentence because it’s a duplicate.

---

## [Decision Letter · Decision Letter 1]

23 Jan 2023

PONE-D-22-33603R1CLINICAL PREDICTORS OF MORTALITY IN PATIENTS WITH PSEUDOMONAS AERUGINOSA INFECTIONPLOS ONE

Dear Dr. Kanafani,

Thank you for submitting your manuscript to PLOS ONE. After careful consideration, we feel that it has merit but does not fully meet PLOS ONE’s publication criteria as it currently stands. Therefore, we invite you to submit a revised version of the manuscript that addresses the points raised during the review process.

We look forward to receiving your revised manuscript.

Kind regards,

Benjamin M. Liu, MBBS, PhD, D(ABMM), MB(ASCP)

Academic Editor

PLOS ONE

Journal Requirements:

Reviewers' comments:

Reviewer's Responses to Questions

**Comments to the Author**

1. If the authors have adequately addressed your comments raised in a previous round of review and you feel that this manuscript is now acceptable for publication, you may indicate that here to bypass the “Comments to the Author” section, enter your conflict of interest statement in the “Confidential to Editor” section, and submit your "Accept" recommendation.

Reviewer #1: All comments have been addressed

Reviewer #2: All comments have been addressed

2. Is the manuscript technically sound, and do the data support the conclusions?

Reviewer #1: Yes

Reviewer #2: Yes

3. Has the statistical analysis been performed appropriately and rigorously? 

Reviewer #1: Yes

Reviewer #2: Yes

4. Have the authors made all data underlying the findings in their manuscript fully available?

Reviewer #1: Yes

Reviewer #2: Yes

5. Is the manuscript presented in an intelligible fashion and written in standard English?

Reviewer #1: Yes

Reviewer #2: Yes

6. Review Comments to the Author

Reviewer #1: (No Response)

Reviewer #2: Methods

Please explain "healthcare-associated" infections in the text as hospital adquired infections are also health care associated - you did it out of the paper, please see answers to reviers

Please note that the place of isolation is not the same of type of infection - we may have a pielonephritis or a simple ITU from a positive urine culture, this should be clear, in all the results

I wonder from the 24 samples out of the most common places of isolation how may cases from cerebrospinal samples were positive, and what was the course of the disease - this is a quite rare situation would be important to detail

Please answer the question about infections acquired in ICU - were they different?

Please correct in

line 79 -cut "Written consents taken"

line 149 ("in" in spite of "is") and 164 (confounders "4")

7. PLOS authors have the option to publish the peer review history of their article (what does this mean?). If published, this will include your full peer review and any attached files.

Reviewer #1: No

Reviewer #2: No

---

## [Author Response · Author response to Decision Letter 1]

8 Feb 2023

Reviewer #2: 

Please explain "healthcare-associated" infections in the text as hospital acquired infections are also health care associated - you did it out of the paper, please see answers to reviewers.

Thank you for pointing this out. We added the following definitions:

Infections were considered community-acquired if they manifested outside the hospital or were diagnosed within 48 hours of admission without any healthcare encounter in the past 30 days.

Hospital-acquired infection was defined as an infection that manifested 48 hours or more after hospital admission or within 7 days of discharge.

Healthcare-associated infection was defined as an infection occurring in patients receiving home and/or ambulatory intravenous therapy, chemotherapy, hemodialysis, wound care, specialized nursing care, or who had attended a hospital clinic within the last 30 days; patients hospitalized in an acute care hospital for ≥ 2 days within the last 90 days; and those residing in a nursing home or long-term care facility.

2.Please note that the place of isolation is not the same of type of infection - we may have a pyelonephritis or a simple ITU from a positive urine culture, this should be clear, in all the results

Thank you, we replaced the “site of infection” phrase with “type of infection” throughout the text. 

3.I wonder from the 24 samples out of the most common places of isolation how many cases from cerebrospinal samples were positive, and what was the course of the disease - this is a quite rare situation would be important to detail.

Thank you for your comment. We had one CSF sample from a lymphoma patient who had a good outcome. We will be happy to add more details to the text if so required.

4.Please answer the question about infections acquired in ICU - were they different? ICU acquired infections were more severe than the others? And how many patients were in the need of ICU admission?

Thank you for your comments.

On bivariable analysis, we found that ICU admission was associated with a worse outcome (mortality). However, this relationship was no longer significant when we adjusted for confounders as evidenced by the regression analysis in Table 4. A total of 53/196 (27%) patients required ICU admission. With a p-value of 0.011, ICU mortality was 17/55 (30.9%) and non-ICU mortality was 21/196 (17.5%). 

We have added this information to the text.

Please correct in

line 79 -cut "Written consents taken" done 

line 149 ("in" in spite of "is") ?? done

 and 164 (confounders "4") done

---

## [Decision Letter · Decision Letter 2]

13 Feb 2023

CLINICAL PREDICTORS OF MORTALITY IN PATIENTS WITH PSEUDOMONAS AERUGINOSA INFECTION

PONE-D-22-33603R2

Dear Dr. Kanafani,

We’re pleased to inform you that your manuscript has been judged scientifically suitable for publication and will be formally accepted for publication once it meets all outstanding technical requirements.

Kind regards,

Benjamin M. Liu, MBBS, PhD, D(ABMM), MB(ASCP)

Academic Editor

PLOS ONE

Additional Editor Comments (optional):

Reviewers' comments:

Reviewer's Responses to Questions

**Comments to the Author**

1. If the authors have adequately addressed your comments raised in a previous round of review and you feel that this manuscript is now acceptable for publication, you may indicate that here to bypass the “Comments to the Author” section, enter your conflict of interest statement in the “Confidential to Editor” section, and submit your "Accept" recommendation.

Reviewer #2: All comments have been addressed

2. Is the manuscript technically sound, and do the data support the conclusions?

Reviewer #2: (No Response)

3. Has the statistical analysis been performed appropriately and rigorously? 

Reviewer #2: (No Response)

4. Have the authors made all data underlying the findings in their manuscript fully available?

Reviewer #2: (No Response)

5. Is the manuscript presented in an intelligible fashion and written in standard English?

Reviewer #2: (No Response)

6. Review Comments to the Author

Reviewer #2: (No Response)

7. PLOS authors have the option to publish the peer review history of their article (what does this mean?). If published, this will include your full peer review and any attached files.

Reviewer #2: No

---

## [Editor Report · Acceptance letter]

20 Apr 2023

PONE-D-22-33603R2 

Clinical predictors of mortality in patients with pseudomonas aeruginosa infection 

Dear Dr. Kanafani:

I'm pleased to inform you that your manuscript has been deemed suitable for publication in PLOS ONE. Congratulations! Your manuscript is now with our production department. 

Kind regards, 

on behalf of

Dr. Benjamin M. Liu 

Academic Editor

PLOS ONE